# Composite Polyurethane-Polylactide (PUR/PLA) Flexible Filaments for 3D Fused Filament Fabrication (FFF) of Antibacterial Wound Dressings for Skin Regeneration

**DOI:** 10.3390/ma14206054

**Published:** 2021-10-13

**Authors:** Paweł Szarlej, Iga Carayon, Przemysław Gnatowski, Marta Glinka, Martyna Mroczyńska, Anna Brillowska-Dąbrowska, Justyna Kucińska-Lipka

**Affiliations:** 1Faculty of Chemistry, Department of Polymer Technology, Gdansk University of Technology, 80-233 Gdansk, Poland; okseren22@gmail.com (P.S.); przemyslaw.gnatowski@pg.edu.pl (P.G.); juskucin@pg.edu.pl (J.K.-L.); 2Faculty of Chemistry, Department of Analytical Chemistry, Gdansk University of Technology, 80-233 Gdansk, Poland; marglink@student.pg.edu.pl; 3Faculty of Chemistry, Department of Molecular Biotechnology and Microbiology, Gdansk University of Technology, 80-233 Gdansk, Poland; martyna.mroczynska@pg.edu.pl

**Keywords:** composite, polyurethane, poly(lactic acid), 3D printing, skin engineering, amikacin sulfate

## Abstract

This paper addresses the potential application of flexible thermoplastic polyurethane (TPU) and poly(lactic acid) (PLA) compositions as a material for the production of antibacterial wound dressings using the Fused Filament Fabrication (FFF) 3D printing method. On the market, there are medical-grade polyurethane filaments available, but few of them have properties required for the fabrication of wound dressings, such as flexibility and antibacterial effects. Thus, research aimed at the production, characterization and modification of filaments based on different TPU/PLA compositions was conducted. The combination of mechanical (tensile, hardness), structural (FTIR), microscopic (optical and SEM), degradation (2 M HCl, 5 M NaOH, and 0.1 M CoCl_2_ in 20% H_2_O_2_) and printability analysis allowed us to select the most promising composition for further antibacterial modification (COMP-7,5PLA). The thermal stability of the chosen antibiotic—amikacin—was tested using processing temperature and HPLC. Two routes were used for the antibacterial modification of the selected filament—post-processing modification (AMI-1) and modification during processing (AMI-2). The antibacterial activity and amikacin release profiles were studied. The postprocessing modification method turned out to be superior and suitable for wound dressing fabrication due to its proven antimicrobial activity against *E. coli*, *P. fluorescens*, *S. aureus* and *S. epidermidis* bacteria.

## 1. Introduction

One of the most common injuries of the body is skin tissue damage, which may be associated with pain, swelling or exudation. Wound healing is a natural biological process that occurs in the wound and leads to its closure and, as a consequence, to the formation of a scar. Chemical reactions and physical phenomena occurring during the regeneration process cause the increase of the skin tissue’s tensile strength [1,2].

Due to wound cleanliness and healing conditions, they could be divided into clean (class 1), clean–contaminated (class 2), contaminated (class 3) and dirty–infected wounds (class 4). The first class includes uninfected injuries with no inflammation, which do not spread into the genital, urinary, digestive and respiratory tracts. Clean–contaminated wounds are slightly contaminated and could affect the digestive, genital, urinary and respiratory tracts in a controlled way. The third group contains contaminated wounds which are open, fresh and may arise from the omission of the sterile wound healing techniques. Class 4 wounds are dirty–infected and very often have an origin in the inaccurate treatment of traumatic wounds. They are also characterized by the presence of devitalized tissue caused by microorganisms [3].

In most cases, skin tissue injuries include wounds from classes 2, 3 and 4. Therefore, appropriate treatment should be instituted to reduce or completely remove discomfort and to protect the wound from further infection as well as from external factors, such as drying and mechanical injuries. Nowadays, many innovative wound healing methods that accelerate the injury regeneration process are available [4]. Some of them are part of the issues of tissue engineering (TE) [5].

The main target of tissue engineering is to obtain a biological material that would be characterized by properties reflecting the characteristics of natural tissue. Material functions would therefore focus on reconstructing fragments or whole damaged tissues or organs [6,7]. Due to the strong need to personalize wound therapy, which depends on the type or size of the patient’s wounds, additive manufacturing (AM), also known as 3D printing (3DP) [8], draws increasing attention [9,10]. AM is a complex process, which includes the design and fabrication of material. The design has to be done using appropriate CAD (Computer Aided Design) software [11] or DICOM (Digital Imaging and Communications in Medicine) files [12]. The main principle of the fabrication of a material using AM is applying subsequent layers of material, until the desired shape is achieved. 3DP allows us to obtain dressings with a designed shape, microarchitecture and geometry. The structure of printed constructs is more repeatable in comparison with traditional methods of production. The biomaterials produced using AM show good cell adhesion and migration, which results in faster skin tissue regeneration [13,14].

Fused Deposition Modeling (FDM)/Fused Filament Fabrication (FFF) is one of the most common 3DP techniques due to its low cost and high availability. FDM is based on the controlled extrusion of thermoplastic filaments. The filament is heated in the printer to a semiliquid state and then the material is extruded through the nozzle and deposited on the printer’s table layer by layer. Successive layers fuse together, thus creating a designed model [15]. 3DP is distinguished by the high precision and reproducibility. Moreover, the FDM technique allows us to obtain pores of almost any shape and size, and does not require the use of solvents or toxic binders [16].

Due to the combination of unique properties and susceptibility to modification, thermoplastic polyurethanes (TPUs) are commonly used polymers in the field of medical applications. They are characterized by high biocompatibility, biodegradability, adequate bending strength and resistance to abrasion, as well as relatively low cytotoxicity [17]. The specific properties of polyurethane (PUR) materials emerge from their segmental structure, substrates used for their synthesis and production methods. PURs synthesized using polyols and diisocyanates contain urethane groups strongly bound by hydrogen interactions. PURs could achieve sufficient physical and mechanical properties for biomedical applications, while maintaining flexibility and hemocompatibility [18,19,20].

PURs also have a wide application in regenerative medicine. They could be used as breast implants, bone adhesives, aortic transplants or heart valves, tissue scaffolds [17] and wound dressings [21]. They are currently synthesized from degradable polyester or polyether polyols forming soft segments (SS), aliphatic diisocyanates and low molecular weight chain extenders, together forming hard segments (HS). This type of material provides the opportunity to control the duration of the in vivo biodegradation process by changing the ratio of polyol, diisocyanate and chain extender. In addition, it has been shown that the degradation products of such polyurethanes are nontoxic and safe for living organisms [20,22,23].

PURs’ great biocompatibility and desirable mechanical properties make them an interesting choice for preparation filaments for FDM [24]. It was found that Tao et al. [25] prepared polyurethane/poly(lactic acid) (PUR/PLA) composite filaments for 3D printed orthesis. Carayon et al. [26] have successfully obtained PUR/PLA foams modified with ciprofloxacin, which had antibacterial properties and possible application in wound treatment. Due to the chemical formula of PLA and PUR, it is possible to obtain compatible blends. Ester groups from PLA are compatible with the soft segments of PUR and could also create hydrogen bonds with the urethane groups from hard segments of PUR. The high compatibility of PUR/PLA blends allows proper extrusion and printability of composite filament [25,27]. PLA filaments are broadly applicated as a feedstock material for FDM 3D printers [28]. PLA is a nontoxic and biodegradable polymer characterized by high mechanical strength, so it could improve the biocompatibility and mechanical properties of the filament made of PUR/PLA composition. Furthermore, PLA based filaments could also be used for the fabrication of biocompatible and biodegradable constructs. For example, Ranjan et al. obtained filaments made of PUR, hydroxyapatite (HA) and chitosan (CS) that could potentially be used for bone cell growth supporting constructs [29].

The current literature does not provide information about other biomedical applications of PUR/PLA composite filaments. Therefore, this paper is focused on the manufacturing of PUR/PLA composite filaments for FDM printing of elastic matrices supporting skin tissue regeneration. The obtained filaments were loaded with amikacin sulphate to obtain antimicrobial properties. The mentioned pharmaceutical substance is a broad-spectrum antibiotic, characterized by the high activity against Gram-positive and Gram-negative bacteria. Abbasi et al. [30] confirmed that hydrogel wound dressings based on sodium alginate containing 1% *w*/*w* of sulphate amikacin exhibited antibacterial properties against *S. aureus* and *P. aeruginosa*. Based on in vivo tests (conducted on Sprague Dawley rats), it was concluded that amikacin sulphate loaded hydrogels show high potential for using as a support for the wound healing process [30].

This paper presents the process of different ratio TPU/PLA composite filaments manufacturing. Hence, the mechanical properties (tensile strength test, hardness), structural analysis (FTIR), microscopic observations (optical microscopy, SEM) and short-term degradation studies were conducted to choose the best composition enabling proper printability using the FDM technique. The chosen TPU/PLA composition was loaded with amikacin sulphate to obtain antimicrobial properties. After modification, the microbiological test and amikacin sulphate releasing profile analysis were conducted to check the potential usefulness of the obtained filaments for skin tissue regeneration applications.

## 2. Materials and Methods

### 2.1. Materials Fabrication

Seven polymeric compositions were fabricated with the use of two homopolymers of synthetic origin: thermoplastic poly(ester urethane) (PUR, Epaline 390A10 25, Epaflex Polyurethanes, Cassolnovo, Italy) and polylactide (PLA, Ingeo7032D, Nature Works, Minnetonka, MN, USA). PUR was chosen due to its excellent flexibility and PLA according to its “fast degradable” characteristic. Polymeric compositions were prepared as follows: polymer pellets of PUR and PLA were mixed manually together at different ratios. In the next step, the prepared compositions were mixed by using an extruder Brabender GNF106/2 within a processing window (185–205 °C, 70–120 rpm) to fabricate filaments (Table 1) of 3.0 ± 0.1 mm diameter, suitable for 3DP FDM technology. Some of the material was then ground and injection molded (Battenfeld HM 45/130 B6E) to obtain samples for tensile tests.

### 2.2. Fourier Transform Infrared Spectroscopy (FTIR)

The FTIR analysis was performed with the use of a Nicolet 8700 Spectrometer in a spectral range from 4000 to 500 cm^−1^, averaging 256 scans with a resolution of 4 cm^−1^.

### 2.3. Tensile Test

Tensile strength (T_Sb_), Young modulus (E), elongation at break (ε) and permanent elongation (ε_p_) of raw polymers (TPU, PLA) and obtained PUR/PLA compositions were studied using the universal testing machine, Zwick & Roell Z020, according to PN-EN ISO 527:2020 and PN-ISO 37:2007 standards with a crosshead speed of 150 mm/min and a test length of 20 mm.

### 2.4. Brittle Fractures

The brittle fractures of the filaments were made by immersing the sample in liquid nitrogen for 30 s and breaking it perpendicularly to the filament axis using pliers.

### 2.5. Hardness

Hardness was measured by using Shore method according to PN-EN ISO 868:2005 using Electronic Shore Type D Durometer. The results were determined as the average of ten measurements. The obtained data were presented with Shore degree (Sh D).

### 2.6. Short Term Degradation Studies in Selected Media

The short-term degradation studies of the obtained filaments were performed in selected media: 2 M HCl, 5 M NaOH, and 0.1 M CoCl_2_ in 20% H_2_O_2_. This is a standard procedure previously reported in the literature [31,32,33,34,35,36]. Filaments were cut into samples of 10 mm length and 3 mm diameter. Prepared samples were dried and weighed in thermobalance (RADWAG MAX50/SX), set at 60 °C. For each used medium, six samples of the studied material were placed in 24-well cell culture plates filled with degradation medium. Samples were then incubated at 37 °C for 15 days. Mass changes of the samples were examined after 15 days in oxidative, acidic, and alkaline media. Sample mass change measurements were as follows: samples were taken out from the container and put into a paper sheet to reduce the medium excess. Samples were then placed in the thermobalance (set at 60 °C), where they were dried to a constant mass and weighed. Mass loss was calculated by Formula (1):(1)$$S=\left(\frac{m_{0}-{\;m}_{1}}{m_{0}}\right)\times 100,$$
where m_1_ is the sample weight after 15 days of incubation (g) and m_0_ is the sample weight before the test (g). The results were the arithmetic mean of six measurements.

### 2.7. Calcification Study of Filament Samples

Golomb and Wagner’s Compound was used to perform the calcification study. The calcification metastable solution consisted of 3.87 mM CaCl_2_, 2.32 mM K_2_HPO_4_, yielding a ratio of calcium to phosphate (Ca/PO_4_) = 1.67, and 0.05 M Tris Buffer (in this study C_4_H_11_NO_3_) dissolved in 1 L of reverse osmosis (RO) water [37,38]. TPU, PLA and COMP-2,5PLA—COMP-12,5PLA filaments were cut into samples of 10 mm length and 3 mm diameter. Prepared samples were dried and weighed in a thermobalance (RADWAG MAX50/SX) set at 60 °C. Then, six samples of each studied PUR material were placed in a 24-well cell culture plate filled with Golomb and Wagner’s Compound at 37 °C. The progress of the calcification was studied by optical microscopy.

### 2.8. Optical Microscopy

Surface of brittle fractures, samples after tensile tests and the progress of calcification of polymeric compositions were studied with an optical microscope XREC at a magnification of ×40 and ×800.

### 2.9. Scanning Electron Microscopy (SEM)

Surfaces of polymeric compositions were studied using FEI Quanta 250 FEG at an accelerating voltage of 10 kV. Samples were covered with a 15 nm layer of gold in a sputter-coater Leica EM SCD 500.

### 2.10. Printability

Prepared filaments were tested for printability using FlashForge Inventor I^®^ (FlashForge, Jinhua, China) 3D FFF printer. Gcode files were prepared using FlashPrint (4.2.0 version) (FlashForge, Jinhua, China) software. For the printed object, cube with the side length of 2 cm was chosen. The following parameters were used: bed temperature; 50 °C, base printing speed: 30 mm/min; layer height: 0.18 mm; infill: 15%, linear; thickness: three contours; closing layers: three top and bottom. Three attempts for each filament were made: one with 210 °C, one with 220 °C and one with 230 °C nozzle temperature.

### 2.11. cAmikacin Thermal Stability

To analyze amikacin thermal stability, a simulation of the extrusion process was performed. The samples of pure amikacin were weighted and maintained in an oven in 195 °C for 10 and 30 min. Next, after cooling to room temperature (5 min, 22 °C), the samples were dissolved with distilled water and analyzed using HPLC-MS/MS.

### 2.12. Filament Loading with Amikacin

For antibacterial modification, one of the compositions was chosen based on physicochemical properties and printability. Drug loading was performed by two routes: immersion and coprocessing. The first route (AMI-1) was treatment of the fabricated filament. Literature reports that amikacin sulphate is stable in water solution for 7 days at 23 °C [39]. Small pieces of the filament (0.06–0.08 g) were immersed in 5 mL of 0.5% amikacin solution in deionized water at 20 °C. The filament was taken out after 7 days of immersion and dried in a vacuum dryer at 50 °C for 12 h and weighed. The second route (AMI-2) was drug loading during fabrication. General procedure was the same as for raw material preparation but with the addition of 0.5 wt% of AMI during the first step. Following that procedure, a filament loaded with AMI was obtained.

### 2.13. Antibacterial Activity

Antibacterial activities of AMI modified filaments were evaluated against *E. coli*, *P. fluorescens*, *S. aureus* and *S. epidermidis* using the disk agar diffusion method. Bacterial strains were obtained from the collection of the Department of Molecular Biotechnology and Microbiology, Gdańsk University of Technology. Stock cultures were maintained by periodic subculture on nutrient agar slants which were stored at 4 °C. Before each experiment, bacterial strains were refreshed by growing in LB broth medium and incubated for 24 h at 37 °C. LB broth medium was prepared by dissolving 10 g NaCl, 10 g peptone, and 5 g yeast extract in 1 L of distilled water and then autoclaved (121 °C, 1.5 atm, 20 min) and cooled to room temperature. All reagents were supplied by BTL Sp. z o.o., Lodz, Poland. Then, the bacterial cultures were diluted tenfold using LB medium, and 0.1 cm^3^ of the bacterial suspensions were spread over the LB agar plates and incubated at 37 °C for 24 h. The filaments were cut, sterilized with 70% ethanol and followed by drying under UV lamp (30 min). The filaments were gently placed on the agar plates using forceps, and the plates were incubated at 37 °C for 24 h. After the incubation, the presence or absence of zones of bacterial growth inhibition around the samples of filaments was checked. Antibacterial activities of the filaments were tested for five samples of the following filaments: COMP-7,5PLA, AMI-1 and AMI-2. During the microbiological test, one piece of filament was placed on the surface of each agar plate.

### 2.14. In Vitro Amikacin Release Profile—Sample Preparation

Stock solution of amikacin (1 mg/mL) was prepared in deionized water and stored in −3 °C. Working standard solutions (2.5, 5, 12.5, 25, 50, 75 µg/mL) were prepared by dissolution of the stock solution with deionized water. After drying, samples of AMI-modified filaments were immersed in 1.5 mL of deionized water in Eppendorf’s vials and left for a specified period of time: 2 min, 5 min, 30 min, 1 h, 3 h, 3 days, 5 days and 7 days at 37 °C in static conditions (without stirring). Each of the samples was made in triplicate. Then, the solution was transferred into the chromatographic vials and analyzed.

### 2.15. In Vitro Amikacin Release Profile—HPLC-MS/MS

HPLC-MS/MS determination of amikacin (Figure 1) in multiple reaction monitoring mode (multiple reaction monitoring—586.4–163.2; declustering potential—126 V; collision energy—47 V) was performed using an Agilent 1200 Series Rapid Resolution LC system (USA) consisting of a binary pump, a high-performance SL autosampler, a thermostated column compartment and an online degasser. The system was coupled to the Q-Trap 4000 triple quadrupole mass spectrometer (AB SCIEX, USA) with an electrospray ionization (ESI) source working in positive ion mode. Other parameters of ESI source were as follows: curtain gas pressure: 25 psi, source temperature: 450 °C, nebulizer gas pressure: 35 psi, heater gas pressure: 45 psi and capillary voltage: 5000 V.

For the determination of AMI amount after release into the water, the fast HPLC-MS/MS method was applied. The total run of a single analysis was 4 min. Mobile phase consisted of water and methanol mixture (80:20 *v*/*v*) with 0.1% formic acid. Flow rate of 0.3 mL/min was used and the injection volume was 3 µL. For separation, Agilent Ecilpse Plus C18 3.5 µm (4.6 × 100 mm) chromatography column was used. The temperature of the column was maintained at 35 °C.

## 3. Results

### 3.1. Fourier Transform Infrared Spectroscopy (FTIR)

The FTIR spectra of the prepared filaments were shown in Figure 2. The spectra of composite filaments (COMP) are similar and there are no clear differences between them. Strong signal at 1750 cm^−1^ could be attached to C=O stretching vibration of ester groups of PLA, and a strong signal around 1690 cm^−1^ could be attached to C=O stretching vibration of urethane group. Medium signal at around 1550 cm^−1^ could be attached to N-H bending vibrations of the urethane group. Weak signals at 3300 cm^−1^ (N-H stretching vibrations of urethane group) and 2900 cm^−1^ (symmetrical and asymmetrical stretching vibrations of C-H bond of -CH_2_-) could be observed. Signals at between 1200 and 1000 cm^−1^ could be associated to the stretching vibration of C-O and C-(C=O)-O of urethane units.

### 3.2. Tensile Test and Brittle Fractures

The tensile test results of the studied polymeric compositions were shown in Figure 3. The tensile strengths of the obtained compositions were similar to neat TPU, and the value oscillates around 30 MPa for nearly all COMP samples. Only one sample (COMP-2,5PLA) showed lower tensile strength (25 MPa). The elongation at break values of COMP-2,5PLA and COMP-5PLA samples were significantly higher than for neat TPU, but the samples with higher amount of PLA had similar elongation at break to neat TPU. The permanent elongation was increasing with the amount of PLA added, up to 7.5%, where it stabilized at around 130% (the permanent elongation of PLA is not shown in figure and it was around 0%). In comparison to the neat TPU samples, the Young modulus decreased for COMP-2,5PLA, and then was increasing up to 40 MPa for samples with 7.5% and above of PLA. High standard deviations could be attributed to the poor homogeneity of materials, which could be observed in brittle fracture photographs (Figure 4 and Appendix A) and in surfaces SEM photographs of the samples (Figure 4). COMP samples after tensile testing showed a similar texture to that of neat TPU samples (Figure 4 and Appendix A). The best samples in terms of tensile tests were the ones with 7.5% of PLA and above. The results are comparable with the literature, as was shown by Jašo et al. [40], 10 wt% addition of PLA into TPU matrix results in an increase in modulus compared to neat TPU without changing other tensile parameters. The literature also reports that the tensile strength of PLA/TPU compositions was similar to TPUs with up to 50% content of PLA [41].

### 3.3. Hardness

The hardness test results are shown in Figure 5. The increasing amount of PLA in composition increases the hardness of the material. All compositions have higher hardness than for neat TPU. The results correspond to previous findings in the literature, where the addition of PLA increases the hardness [40].

### 3.4. Short Term Degradation Studies in Selected Media

In Figure 6, short term degradation studies were presented. Neat PLA degraded completely in NaOH after 15 days. With an increasing amount of PLA in composition, the increasing degradation in NaOH could be observed in materials. A similar trend could be observed for samples after degradation in HCl, but for all compositions, the degradation in HCl was slower than for neat TPU. Degradation in CoCl_2_ in H_2_O_2_ was negligible. In Figure 7, the spectra of COMP-7,5PLA before and after degradation studies were presented. Similar figures for other compositions could be found in Appendix A. It could be said that the degradation in HCl mostly affects the urethane part of the composition—the disappearance of the N-H signal at 3300 cm^−1^, C=O signal at 1690 cm^−1^ and loss of intensity of N-H signal at 1550 cm^−1^. It could be also said that degradation in NaOH mostly affects the lactide part of composition—the disappearance of the C=O ester signal at 1750 cm^−1^. The analysis of the spectra of the sample after degradation studies in CoCl_2_ in H_2_O_2_ proves that the degradation was negligible. The photos of the samples before and after degradation in NaOH were shown in Figure 6. The literature reports the susceptibility of PLA to hydrolytic degradation in an alkali environment [42] and an ester-based TPU to acidic degradation [43]. The literature also shows that the increase of PLA content in the composition increases the susceptibility of polymeric compositions to hydrolytic degradation in an alkali environment [41].

### 3.5. Calcification Study of Filament Samples

Figure 8 shows the results of calcification studies. The mass changes were negligible (<0.1%) for nearly all samples. COMP-12,5-PLA has a 0.45% mass increase after calcification. The photos of the samples before and after calcification studies can be found in Figure 9. For samples with 7.5 wt% PLA and above, some salt crystals on the surface of filament could be observed. The lack of mass increase could be explained by the slow hydrolysis of filaments [44], which compensates for the increase from growing salt crystals.

### 3.6. Scanning Electron Microscopy (SEM)

The SEM micrographs taken on the surface of neat samples (Figure 10) show homogeneity issues of compositions, spotted also using optical microscopy (Figure 4 and Appendix A). Agglomerates could be found on surfaces of the samples.

### 3.7. Printability

Three attempts at printing for each filament were made: one with 210 °C, one with 220 °C and one with 230 °C nozzle temperature were done. All compositions up to 7.5 wt% of PLA have proven their printability. The best printing results were observed with a 220 °C nozzle temperature. It was observed that filaments with a high amount of PLA (10 and 12.5 wt%) were clogging the nozzle of the 3D printer, which made those compositions unprintable. Such behavior could be caused by low homogeneity of those samples. In Figure 11, the produced COMP-7,5PLA filament and printed cube are presented.

### 3.8. Filament Loading with Amikacin

Because of the composition, COMP-7,5PLA provided a good balance between mechanical properties, degradability and printability, two amikacin loaded filaments based on this composition were prepared using the immersion (AMI-1) and coprocessing (AMI-2) methods.

### 3.9. Amikacin Thermal Stability

The samples after thermal stability test were shown in Figure 12. The decrease of the amikacin mass was 1.5 ± 0.3% after 10 min of heat treatment and 21 ± 2% after 30 min. It could be said that amikacin is stable at short exposures to high temperatures. Additionally, samples were analyzed using MS-scan mode and the formation of amikacin degradation products was not observed (based on information about amikacin impurities from European Pharmacopoeia [45]). The literature supports the results, according to which the thermal degradation of amikacin starts at around 230 °C [46].

### 3.10. Antibacterial Activity

The results of microbiology studies are summarized in Table 2. The inhibition zones of AMI-1 samples are shown in Figure 13. AMI-1 filament (modification of the fabricated filament) has proven antibacterial activity for all tested samples. Filament AMI-2 (co-extrusion with amikacin) showed antibacterial activity only in one *E. coli* sample, so it has not proven antibacterial activity. It is known that to successfully interact with the environment, the antibiotics have to be present on the surface of the material [47]. Results suggest that, for the co-extrusion fabrication method, a higher amount of amikacin should be used to obtain the antibacterial properties similar to those of the modified filament.

### 3.11. HPLC Test Conditions

The validation of the HPLC-MS/MS method was performed. It consisted of the estimation of linearity, limits of detection (LOD) and limits of quantification (LOQ). The obtained calibration curves were linear within the studied concentration range (Figure 14).

Limits of detection, calculated as LOD = calibration curve intercept standard deviation multiplied by 3.3 and divided by the slope of the calibration curve and limits of quantitation LOQ = 3 × LOD were established: LOD = 0.72 μg/mL and LOQ = 2.2 μg/mL (Table 3).

In addition, the obtained results for samples and standards were also compared using an analytical procedure with fluorenylmethyloxycarbonyl chloride (FMOC-Cl) pre-column derivatization and determination with HPLC with a fluorescence detector (HPLC-FLD) system. Results from both methods were statistically compared (F-Snedecor test, *t*-Student test with α = 0.05). Considering the precision and accuracy, both methods have no statistical differences. The main difference between methods was the total time of sample analysis, which for HPLC-FLD was 80 min and for HPLC-MS/MS the time was only 4 min. In further investigation, the HPLC-MS/MS method was used.

Considered the filament AMI-1 (immersion in amikacin water solution), the release of amikacin has already occurred after 2 min (192 ± 41 μg/mL, Figure 15). This is a consequence of the weak interaction of amikacin with the functional groups of PUR-PLA. With the prolongation of the exposure time, no significant differences in the amikacin concentration were noted (obtained concentration levels after the immersion test were in the range 192–229 µg of AMI in relation to 1 g of dry mass of filament). To determine the time of full amikacin release, samples after 2 and 5 min of immersion were dried and immersed in deionized water for 3 days. After that, solutions were analyzed using HPLC-MS/MS. It was found that the AMI concentrations were below the range of the calibration curve, therefore it confirmed that after 2 min of immersion the total amount of AMI was released from the filament into the water solution.

However, the process of filament preparation was not fully repeatable. It is evidenced by the standard deviation values, calculated for three samples of filaments (Figure 15, Table 4). The high SD values were probably a consequence of amikacin dispersion in a polymer matrix.

In the case of filaments with the composition AMI-2 (co-extrusion with amikacin), the release of amikacin to the aqueous solution after the immersion test was not observed. Even after seven days of filament immersion, the concentration of AMI was below the range of the calibration curve. In addition, the samples were analyzed in the SCAN mode and non-signal of amikacin was observed. Figure 16 shows the differences in the intensity of the amikacin peaks depending on the filament type.

The difference in amikacin release is related to the method used for filament preparation. As was previously proven, amikacin is thermally stable at processing temperature (Figure 12), thus the low amikacin release is probably an effect of the hinderance of antibiotic in polymer matrix. The directly mixing of granulates and active substance in powdered form could cause the immobilization (entrance) of the antibiotic particles inside the filament structure which cannot degrade in hydrolytic conditions (e.g., too small carrier pores, sintering with polymer structure at high temperature treatment during preparation). Additionally, another possible reason is the drug form used in the loading step. It was proven that aminoglycosides used in the solid form cannot interact with polymer functional groups [48]. The usage of the immersion method allows the dissolving of amikacin, which enables the interactions between the drug and polymer functional groups on the surface of the filament.

## 4. Conclusions

It was proven that the compositions of TPU and PLA could be utilized for 3D printing. Generally, the higher amount of PLA in the composition was, the better the properties obtained—higher Young’s modulus, higher hardness and better biodegradability. However, microscopy and printability tests revealed that the obtained samples were less homogenous with higher PLA concentrations. This is a common problem with the one-screw extruder [49]. The homogeneity of the samples could be improved by using large scale equipment, such as twin-screw extruders [50]. Two routes of filament modification with amikacin showed diametrically different behaviors. The first route of post-processing modification resulted in a material with a rapid amikacin release profile. This material could be used in wound dressings, where fast drug delivery is desired. The second route of coextrusion resulted in material with very slow amikacin release. We have proven that amikacin is stable during extrusion at elevated temperatures. This route, if combined with a material with higher biodegradability properties, could be used for the preparation of short-term implants, where drug release should be stable during the implantation time. The amount of amikacin could also be raised to increase the presence of the active agent on the surface of the implant. Such implants could have applications as bone and cartilage (e.g., septum implants) scaffolds.

## Figures and Tables

**Figure 1 materials-14-06054-f001:**
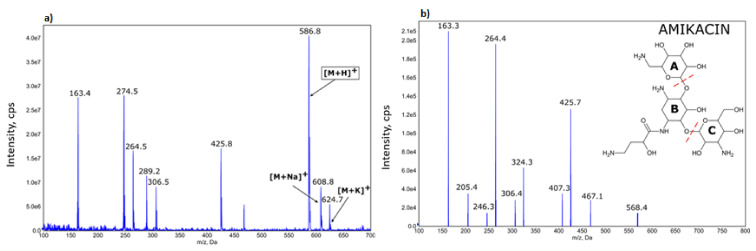
Mass spectrum of amikacin sample (**a**)—ion 586.8 *m*/*z* represents protonated amikacin and was used in quantitative analysis. Mass spectral fragmentation of amikacin obtained using ESI-MS/MS, positive mode (**b**).

**Figure 2 materials-14-06054-f002:**
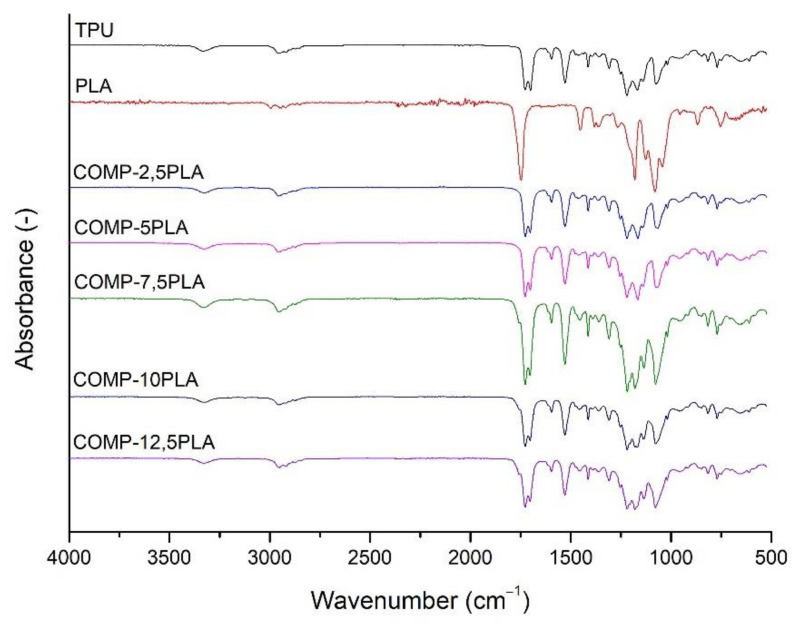
FTIR spectra of the prepared filaments.

**Figure 3 materials-14-06054-f003:**
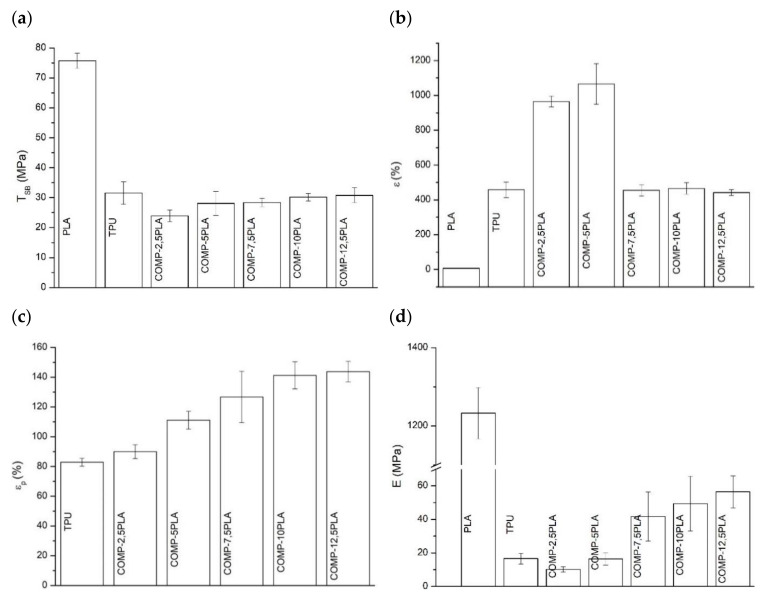
Tensile test results (**a**) tensile strength (T_Sb_), (**b**) elongation at break (ε), (**c**) permanent elongation (ε_p_) of the obtained compositions and (**d**) Young’s modulus (E).

**Figure 4 materials-14-06054-f004:**
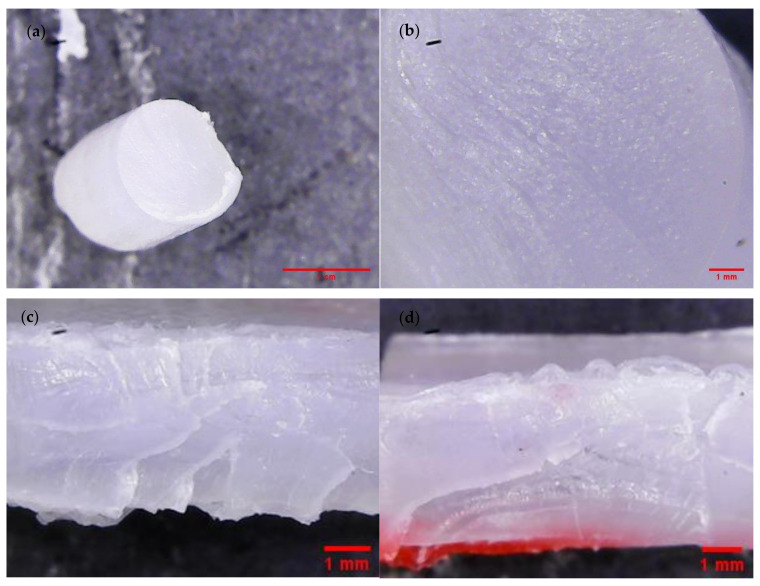
COMP-7,5PLA sample failure mode micrographs: (**a**,**b**) brittle fracture; (**c**,**d**) tensile rupture.

**Figure 5 materials-14-06054-f005:**
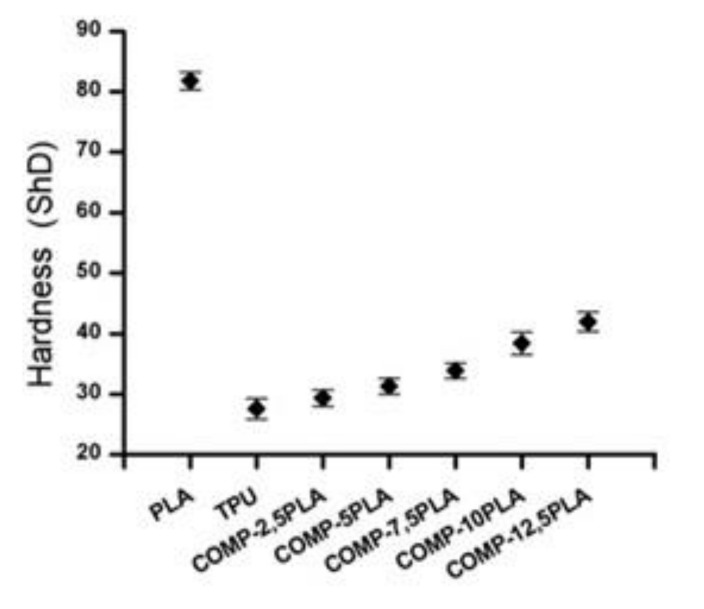
Hardness of obtained compositions.

**Figure 6 materials-14-06054-f006:**
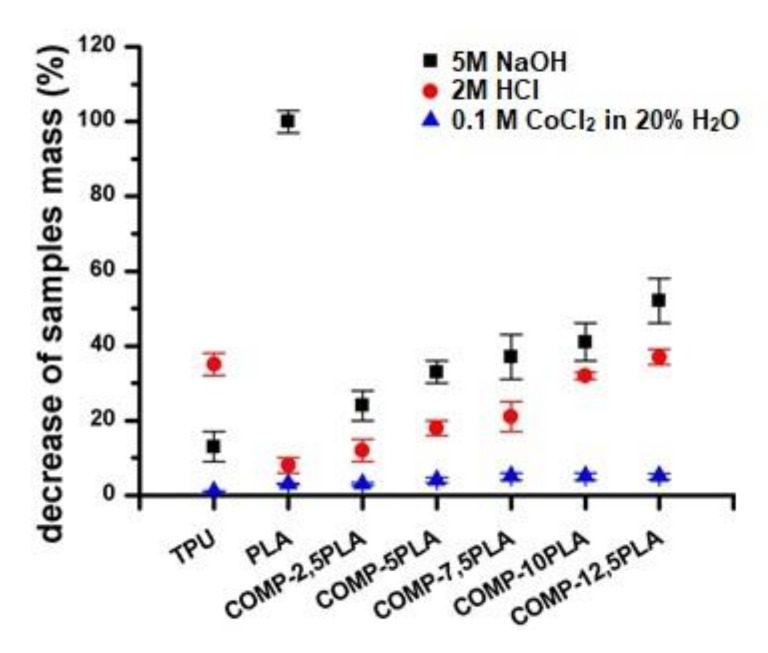
Decrease of samples mass after 15 days of degradation in selected media (0.1 M CoCl_2_ in 20% H_2_O_2_, 2 M HCl and 5 M NaOH).

**Figure 7 materials-14-06054-f007:**
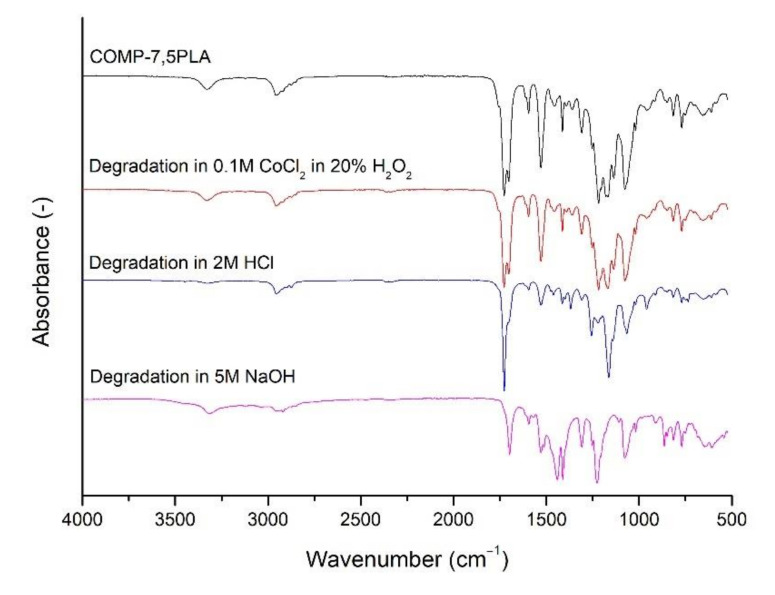
Comparative FTIR spectra of COMP-7,5 PLA sample before and after 15 days of degradation in selected media (0.1 M CoCl_2_ in 20% H_2_O_2_, 2 M HCl and 5 M NaOH).

**Figure 8 materials-14-06054-f008:**
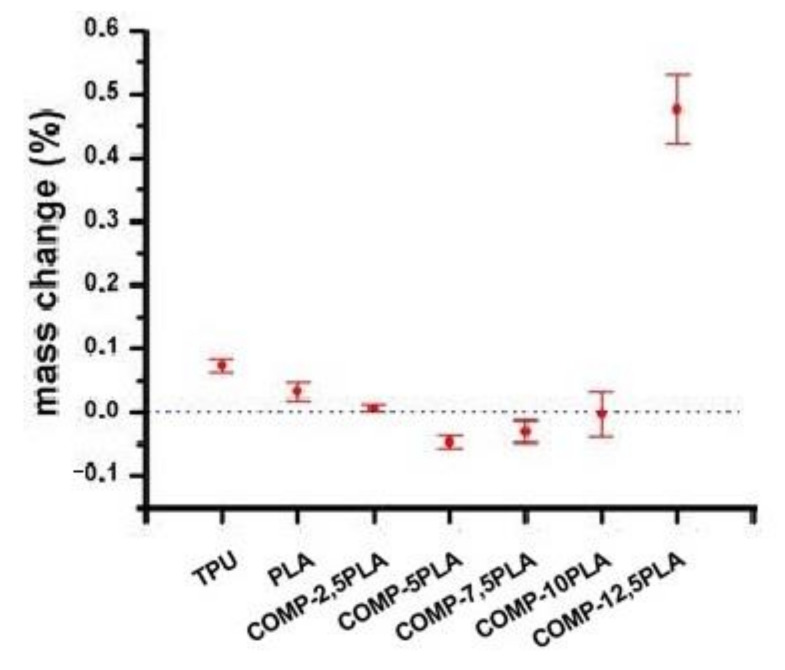
Sample’s mass change in calcification studies.

**Figure 9 materials-14-06054-f009:**
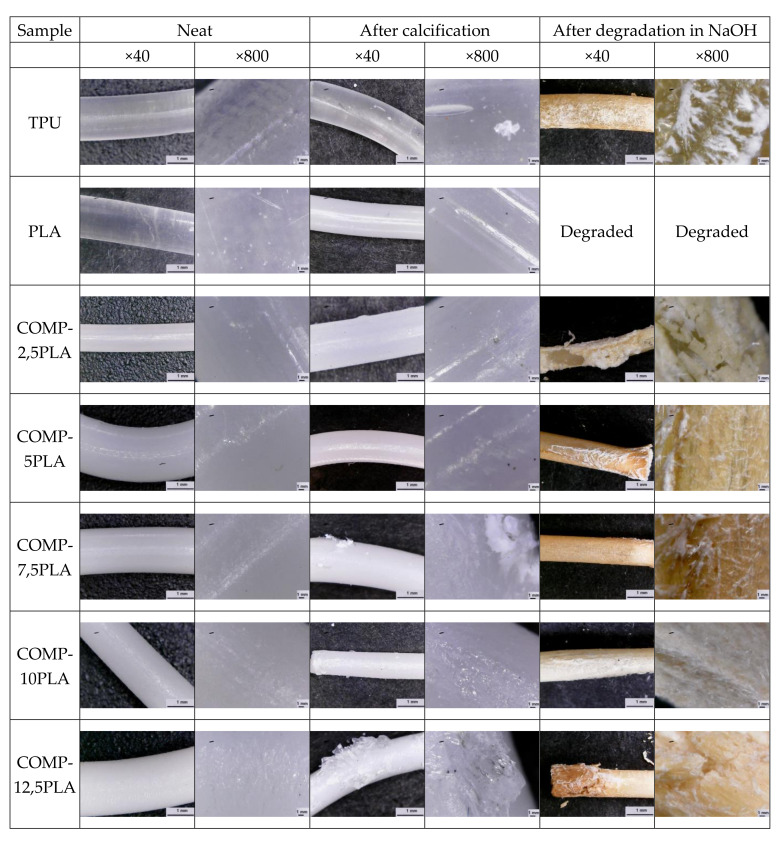
Micrographs of neat samples and after calcification and degradation in NaOH.

**Figure 10 materials-14-06054-f010:**
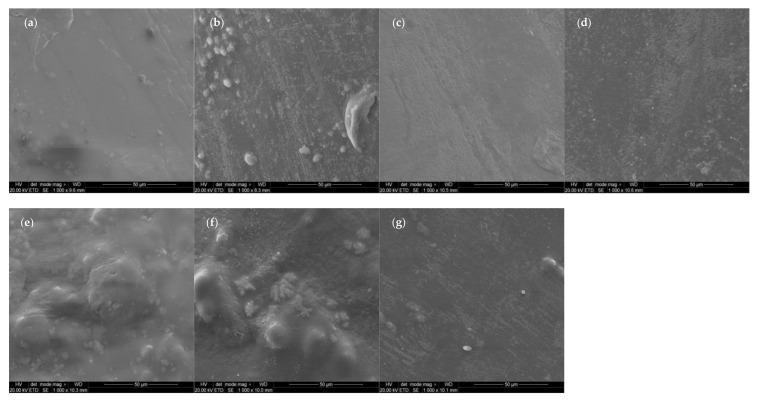
SEM micrographs of surfaces of neat samples: (**a**) PLA, (**b**) TPU, (**c**) COMP-2,5PLA, (**d**) COMP-5PLA, (**e**) COMP-7,5PLA, (**f**) COMP-10PLA, (**g**) COMP-12,5PLA.

**Figure 11 materials-14-06054-f011:**
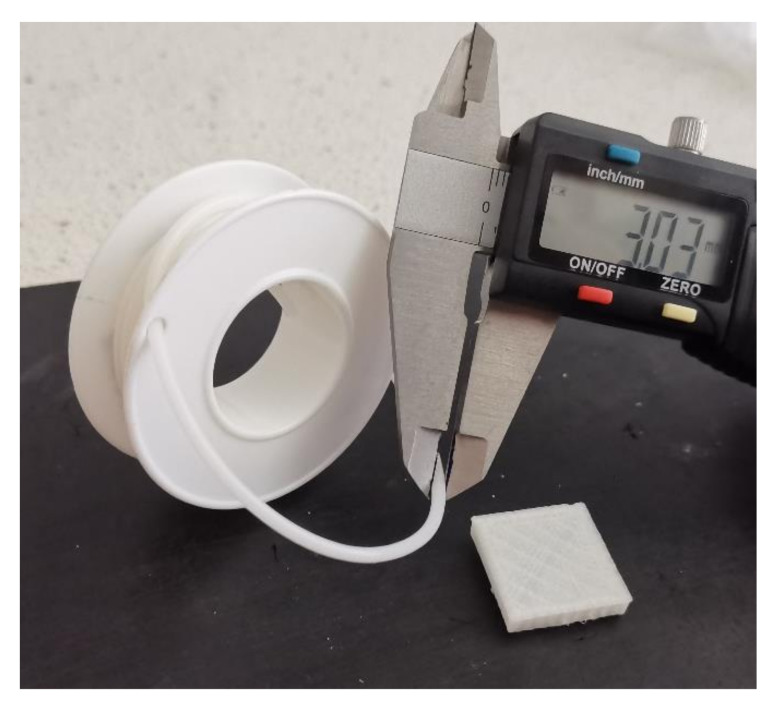
Photography of the produced COMP-7,5PLA filament and printed cube.

**Figure 12 materials-14-06054-f012:**
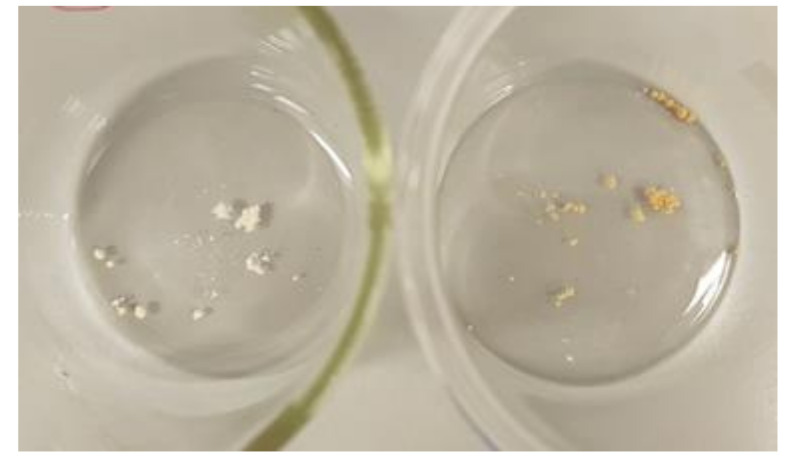
Photo of amikacin samples after heat treatment, after 10 min (**left**) and 30 min (**right**) at 195 °C.

**Figure 13 materials-14-06054-f013:**
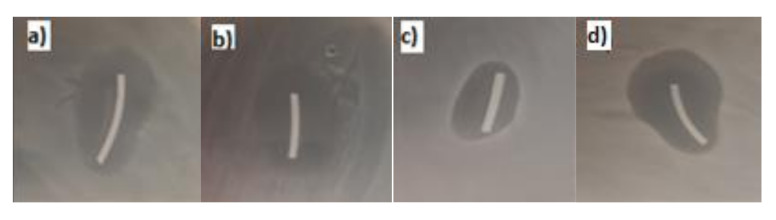
Selected photographs of inhibition zones of AMI-1 samples in bacteria growth (**a**) *E. coli* (**b**) *P. fluorescens* (**c**) *S. aureus* (**d**) *S. epidermidis*.

**Figure 14 materials-14-06054-f014:**
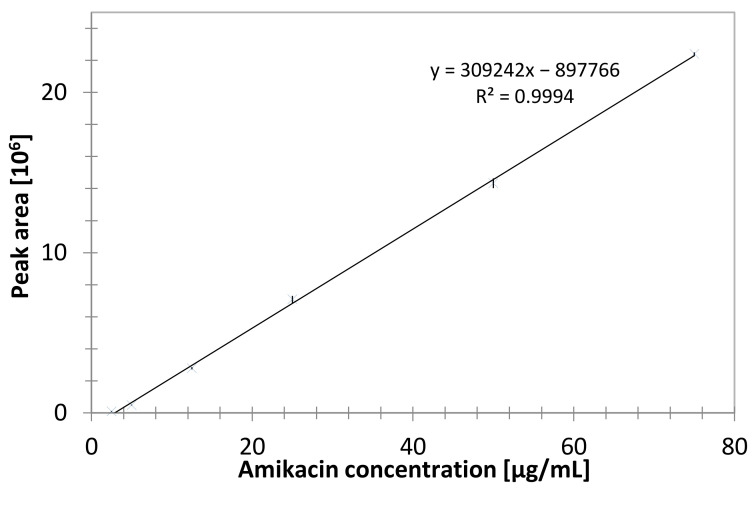
Calibration curve of amikacin (2.5–75 µg/mL).

**Figure 15 materials-14-06054-f015:**
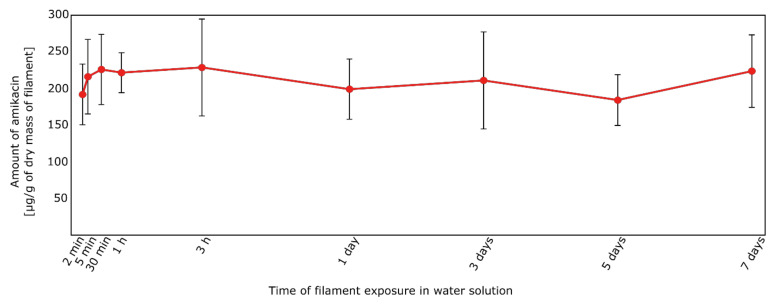
Amount of amikacin [µg/g of dry mass of filament] released from the AMI-1 filaments.

**Figure 16 materials-14-06054-f016:**
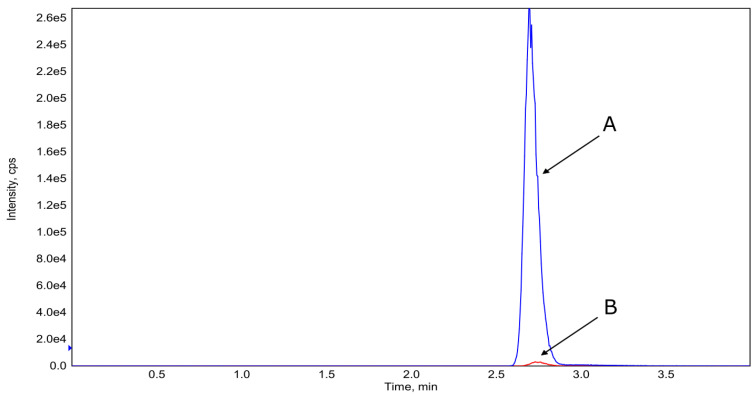
Overlay of HPLC-MS/MS amikacin chromatograms. A—AMI-1 filament after 7 days of immersion in water, B—AMI-2 filament after 7 days of immersion in water.

**Table 1 materials-14-06054-t001:** Symbols of the obtained filaments with a brief explanation and materials’ ratios used for their fabrication.

Filament Symbol	Ratio PUR:PLA	Description
TPU	1:0	Pure PUR
PLA	0:1	Pure PLA
COMP-2,5PLA	39:1	COMP-1 consists of 39 parts of PUR and 1 part of PLA.
COMP-5PLA	19:1	COMP-2 consists of 19 parts of PUR and 1 part of PLA.
COMP-7,5PLA	12:1	COMP-3 consists of 12 parts of PUR and 1 part of PLA.
COMP-10PLA	9:1	COMP-4 consists of 9 parts of PUR and 1 part of PLA.
COMP-12,5PLA	7:1	COMP-5 consists of 7 parts of PUR and 1 part of PLA.

**Table 2 materials-14-06054-t002:** Antibacterial activity tests summary.

Sample	*E. coli*	*P. fluorescens*	*S. aureus*	*S. epidermidis*
COMP-7,5 PLA-1	- *	-	-	-
COMP-7,5 PLA-2	-	-	-	-
COMP-7,5 PLA-3	-	-	-	-
COMP-7,5 PLA-4	-	-	-	-
COMP-7,5 PLA-5	-	-	-	-
AMI-1-1	+ **	+	+	+
AMI-1-2	+	+	+	+
AMI-1-3	+	+	+	+
AMI-1-4	+	+	+	+
AMI-1-5	+	+	+	+
AMI-2-1	-	-	-	-
AMI-2-2	+	-	-	-
AMI-2-3	-	-	-	-
AMI-2-4	-	-	-	-
AMI-2-5	-	-	-	-

* bacteria growth; ** inhibition of bacteria growth.

**Table 3 materials-14-06054-t003:** HPLC calibration parameters.

Calibration Curve Range [µg/mL]	Calibration Curve y = ax + b	S_a_	S_b_	R^2^	LOD [µg/mL]	LOQ [µg/mL]
2.5–75	y = 309,242x − 897,766	1937	67,783	0.9994	0.72	2.2

S_a_—standard deviation of slope of calibration curve, S_b_—standard deviation of intercept of calibration curve, R^2^—coefficient of determination.

**Table 4 materials-14-06054-t004:** Amount of amikacin (after its release from filament AMI-1 into water solution).

Exposure Time	AMI Amount ± SD * [µg/g of Dry Mass of Filament]	CV **
2 min	192 ± 41	22%
5 min	216 ± 51	23%
30 min	226 ± 48	21%
1 h	222 ± 27	12%
3 h	229 ± 66	29%
1 day	199 ± 41	21%
3 days	211 ± 66	31%
5 days	184 ± 35	19%
7 days	224 ± 49	22%

* SD—standard deviation, calculated for 3 prepared filaments; ** CV—coefficient of variation (CV = standard deviation [SD]/mean).

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
