# Peer review of "Composite Polyurethane-Polylactide (PUR/PLA) Flexible Filaments for 3D Fused Filament Fabrication (FFF) of Antibacterial Wound Dressings for Skin Regeneration"

_materials, 2021, doi:10.3390/ma14206054_

Round 1

Reviewer 1 Report

I. Aspects of writing form:

1. The idiomatic wording of sentences and phrases should be checked for structural correctness. Some of them have been highlighted in the document.

2. The writing text should be reviewed, for check inaccuracies of terms, use of units in the graphs.

3. It could be important to clarify the times involved in some test results, for example for micrograph’s after immersion in a specific medium.

4. There are terms that are used, and they are not the most adequate to define scientifically or technically what they are wanting to describe. Some of these terms have been highlighted and some suggestions have been made. In some cases, it would be worth justifying its use.

5. The use of some terms is evidenced, and they are not previously cited as acronyms.

6. The use of scientific terms (for example orthesis) in "introduction" must be verified and the scientific support to the purpose of the research must be validated.

II. Bibliographic references:

1. The bibliography and citations should be reviewed: in some cases what was supported by the citation was not possible to verify, as was the case with the use of the solution of cobalt chloride in hydrogen peroxide.

2. The bibliographic references are not homogeneously cited. References segment is attached indicating some review elements.

Author Response

Thank you for your reviews. Corrections you ma find in the manuscript and our responses you will find in attached document.

Reviewer 2 Report

The paper matches the main aim of the Journal illustrating a technology development in tissue engineering for skin tissue regeneration applications. The structure and properties of the selected materials are sufficiently discussed supporting the originality of the research study.

In particular, the authors propose the application of flexible thermoplastic polyurethane and polylactic acid compositions to produce antibacterial wound dressings by using the Fused Filament Fabrication (FFF) 3D printing method. A modification in these filaments based on TPU/PLA compositions is advanced. Through the analysis performed in the study the best composition for antibacterial modification (COMP-7,5PLA) was first identified and modified with amikacin sulphate so to be tested for thermal stability studying antibacterial activity and amikacin release profiles. Between the two indicated ways ‐- post-processing modification (AMI-1) and modification during processing (AMI2) -- the post processing one was found to be suitable for wound dressings fabrication.

The followed procedure is described in detail in the paper as well as the experimental results that are largely provided.

The paper is interesting for the promising results obtained in the field and further applications. Only few minor suggestions are here indicated:

- In the Introduction (after the first section pag. 3) should be interesting to mention the inclusion of fillers in filament. This, in fact, causes shrinkage which makes printing more accurate. The filaments thus developed exhibited superior properties compared to that of the neat polymer. Important indications of this are given in the following manuscripts (which can be cited):

* Aida, H. J., Nadlene, R., Mastura, M. T., Yusriah, L., Sivakumar, D., & Ilyas, R. A. (2021). Natural fibre filament for Fused Deposition Modelling (FDM): a review. International Journal of Sustainable Engineering, 1-21.

** Calì, M., Pascoletti, G., Gaeta, M., Milazzo, G., & Ambu, R. (2020). A new generation of bio-composite thermoplastic filaments for a more sustainable design of parts manufactured by FDM. Applied Sciences, 10(17), 5852.

- The sections after the Introduction should be numbered (2. Material fabrication; 3. Results). Please correct them.

Also the relative subsections should be numbered or differently the authors could value to change their presentation without a strict distinction of subsection.

- The insertion of Letters for the figures should be homogeneous (see fig. 1 A, B and fig. 3 a,b,c...).

- A revision of tables and figures is required as far as margins and dimensions are concerned.

- As far as the English language is concerned, the paper is written clearly even though its fluency sometimes can be improved. Check the use of definite and indefinite article, especially in section 3.

Here below you can also find some sentences, as an example, that need correction.

Abstract

- This paper addresses the potential application of flexible thermoplastic polyurethane (TPU) and poly(lactic acid) (PLA) compositions as a material for production antibacterial wound dressings via Fused Filament Fabrication (FFF) 3D printing method.

- On the market there are medical-grade polyurethane filaments, but just few of them represents flexibility and antibacterial effect, what is suitable for fabrication of wound dressings.

- Thus, research aimed at production, characterization and modification of filament based on TPU/PLA compositions were undertaken.

Introduction

- Class 4 wounds are dirty – infected, what very often results from inaccurate treatment of traumatic wounds.

- Nowadays, there are lot of innovative wound healing methods that accelerate the injury regeneration process

- Due to the strong need to personalize the therapy depending on the type or size of the patient's wounds

- AM works by the applying subsequent layers of material,

- The structure of printed constructs is more repeatable in comparison with traditional methods of production and also show good cells adhesion and migration, which leads to faster skin tissue regeneration

- .....what results in good physical and mechanical properties, while maintaining flexibility and hemocompatibility, what is very beneficial for biomedical applications

- Due to the PURs great biocompability and broad desirable mechanical properties they can be successfully used for preparation filaments for FDM

- For example, Ranjan et al. obtained filaments made of PUR, hydroxyapatite (HA) and chitosan (CS) that can be potentially use for bone cells growth supporting constructs

Section 2

- Polymeric compositions were obtained as follow:

- Tensile strength (TSb), Young modulus (E), elongaion at break (εb) and permanent elongation (εp) of raw polymers (TPU, PLA) and obtained PUR/PLA compositions were studied by using the universal testing machine Zwick & Roell Z020 according to PN-EN ISO 527-1-3:1998, PN-EN ISO 527-4-52000 and PN-ISO 37:2007 standard with a crosshead speed of 150 mm/min and measuring path of 20 mm.

- Six samples of each studied PUR materials were then placed in 24-well cell culture plates filled with selected media:

Section 3

- The FTIR spectra of prepared filaments are shown on Fig. 2.

- The tensile tests results of studied polymeric compositions are shown on Fig. 3.

- Literature also reports, that tensile strength of PLA/TPU compositions are similar to TPU`s up to 50% content of PLA[35].

- The hardness test results are shown on Fig. 5.

- On Fig. 6 there are presented short term degradation studies.

- Similar trend can be observed for degradation in HCl,

- On Fig. 7 the spectra of COMP-7,5PLA before and after degradation studies can be observed.

- The mass changes were negligible for nearly all samples – bellow 0.1% change.

- The SEM studies (Fig. 10) shows homogeneity issues of samples, spotted also using optical microscopy.

- On Fig. 11 produced filament and printed cube are shown.

- It is known, that the most important place for antibiotics to be, is the surface of material

- It was proven, that the both methods have not statistical differences considered the precision and accuracy.

- But in the case of derivatization method the total time of sample analysis (including sample preparation step) was approx. 80 minutes, but in the case of HPLC-MS/MS method the time was only 4 min.

- In addition, samples after 2 and 5 minutes of immersion were re-tested and immersed for a further 3 days.

- This is probably the result of the type of method used to filament preparation.

Author Response

(The authors gave the same response as above.)

Round 2

Reviewer 1 Report

It is important to review some aspects of form and English Language style and verify if some experimental facts must be included. 

I send in the attached file of the document the observations of aspects that I consider should be reviewed.

Author Response

Dear Reviewers,

Thank you for your valuable notes. We hope that with this improvements our article will be accepted for the publication in Materials.

Regards,

Iga Carayon
